# Earthquake-Induced Waste Repurposing: A Sustainable Solution for Post-Earthquake Debris Management in Urban Construction

Nurullah Bektaş [1],[*] and Maysam Shmlls [2]

1   Department of Structural Engineering and Geotechnics, Széchenyi István University, 9026 Győr, Hungary
2   Department of Architecture and Building Construction, Széchenyi István University, 9026 Győr, Hungary; maysamshmlls@gmail.com
*   Correspondence: nurullahbektas@hotmail.com

**Abstract:** Product sustainability has moved beyond being an elective preference to becoming a certain necessity. However, earthquakes in different regions, particularly Türkiye–Syria, Afghanistan, and Morocco, have produced a substantial amount of construction waste and debris. In the context of green urban initiatives and environmental preservation, theeffective management and reduction of environmental impact (EI) are imperative. This urgency underscores the significance of the study's focus on a ten-story reinforced concrete (RC) dormitory building in Győr, Hungary, chosen as a case study. The research delves into the incorporation of three distinct concrete compositions through seismic design, aligning with the innovative approach of emphasizing recycled aggregate-based concrete to mitigate the EI. Utilizing AxisVM X7 and Revit software, the study meticulously created and analyzed a detailed building model, revealing a significant percentage (35%) and amount (1519.89 tons) of concrete waste that could be incorporated into construction. The results also showed a reduction in both total carbon emissions and the price of materials by falling 27.5% and 9.13%, respectively. We propose an eco-friendly way to effectively reuse debris from earthquakes, focusing on the case study of the 2023 Türkiye–Syria earthquake and encouraging resource efficiency while also addressing the construction waste problems that arise after an earthquake.

**Keywords:** earthquake; buildings; sustainability; recycled concrete aggregate; environment preservation; debris; carbon emission

## 1. Introduction

Recent earthquakes, including the Türkiye–Syria earthquake on 6 February 2023 [1], the Herat Afghanistan earthquake on 11 October 2023 [2], the Marrakesh-Safi Morocco earthquake on 9 September 2023 [3], and the Noto Peninsula Japan earthquake on 1 January 2024 [4], have highlighted the profound vulnerability of existing buildings to the devastating impact of earthquakes. High-impact seismic activity can cause fatalities and extensive structural damage. For instance, the seismic event that struck Türkiye and Syria resulted in the destruction or serious damage of over 156,000 buildings [5], emphasizing the critical necessity of comprehensive risk mitigation strategies. Moreover, after this earthquake left an estimated total debris amount ranging between 116 and 210 million tons [5], highlighting the need of using sustainable waste management techniques, it is critical to focus on the environmental implications of such large-spread debris accumulation. The debris consists of a diverse range of materials, e.g., concrete, brick, crushed aggregate, and pieces of steel components, etc. Attention should be drawn to the fact that concrete, which constituted 28.6% of the total demolition waste (33 to 60 million tons of concrete) in the Türkiye–Syria earthquake and established itself as the second most prevalent component after brick, played a significant role in the context of earthquake-induced waste [6]. Since concrete is the second most used material in the world and emits up to 2.8 billion tons of carbon dioxide annually [7], 50% of climate change, 40% of energy consumption, and 50%

of landfill waste [8,9], this discovery emphasizes both the structural effects of earthquakes and their environmental consequences. By assessing the environmental carbon emissions of the materials used in building construction, life cycle assessment (LCA) is required to address the twin challenges of structural fragility and EI [8]. Therefore, to become more resilient and environmentally friendly, the construction industry must develop sustainable waste management practices and alternative materials capable of reducing the impact of debris on humans and the natural environment.

Debris management poses significant challenges to urban and rural society, impacting both economic and social aspects. One major hurdle is the incomplete data available on the amount and composition of the debris, as well as the social and economic factors involved [10]. Moreover, starting to collect data even before disasters occur, not just during the reconnaissance and mitigation phases (short and long term), is essential. This lack of clarity necessitates extensive data analysis. Efficient methods for determining the amount and characteristics of debris are crucial for effective waste allocation, repurposing, and reuse. Another crucial aspect to consider is the transportation of disaster waste for management, debris collection logistics, and determining the optimal network for its movement [11]. However, the limited budget and personnel available for waste management exacerbate these challenges. The lack of comprehensive disaster management emergency plans, combined with the need for immediate debris removal, often results in serious oversights and errors during plan preparation and execution [12].

Nonetheless, the pursuit of sustainability in the concrete sector begins with mitigating the adverse effects of cement production, a major contributor to $CO_2$ emissions. This reduction is crucial to control environmental damage and address the contribution to global warming [13]. Therefore, introducing supplementary cementitious materials or other waste products into concrete presents an opportunity to stabilize this effect, save energy, and conserve natural resources. The research by Jin and Chen [14] indicates that the substitution of fly ash for cement resulted in a 0.7% decrease in energy consumption during the cement production process. Additionally, by integrating non-traditional components such as recycled concrete aggregate as replacements for natural aggregate [15], it becomes viable to create concrete that not only reduces its EI, but also proves more cost-effective than standard concrete. In the technical aspect, choosing the correct percentage of recycled concrete aggregate (in place of natural aggregate) with the best ratio of supplementary cementitious materials (in place of cement) can help improve compressive strength and other mechanical properties [16]. For instance, incorporating fly ash and silica fume into RC can enhance both its mechanical and durability characteristics. Fly ash contributes to improved workability [17], while silica fume enhances the microstructure of recycled aggregate concrete due to its fine particle size and expansive surface area. According to Kou and Poon [17], experimenting with various amounts of fly ash as a cement replacement revealed that 25% fly ash yielded optimal results in the production of recycled aggregate concrete. Additionally, Abed et al. [18] found that utilizing around 12% silica fume enhanced the strength of recycled aggregate concrete. However, some investigations showed a significant reduction in both compressive and flexural strength up to 55% [19].

Utilizing Building Information Modeling (BIM) involves a sophisticated approach reliant on model-based techniques to aid engineers throughout various stages of construction projects, from planning and design to construction and management, enhancing efficiency. Furthermore, BIM contributes to sustainable design efforts by minimizing project expenses, material use, waste, and environmental impact through effective site and logistics management [20]. However, the literature review reveals a significant use of BIM in aiding engineers during the design phase, while LCA has been employed to evaluate the EI of construction materials. Despite these advances, there exists a notable gap in the integration between BIM, structural analysis, and LCA for evaluating structural materials according to various sustainable criteria.

This study introduces an innovative approach, using three types of concrete from the literature for analyzing designed models (dormitory building located in Győr, Hungary—

an RC moment frame structure) by incorporating Revit software [21] for architectural design and AxisVM [22] for structural design, along with LCA. The primary goals of this study are the following: (1) To explore new avenues for sustainable construction, highlighting the potential for a significant integration of crushed concrete, which leads to substantial reductions in carbon emissions and offers an optimal solution for waste management. (2) Additionally, to determine the amount of concrete use while exploring the feasibility of repurposing earthquake-induced concrete waste, with a particular focus on its application in the construction of RC buildings. (3) Furthermore, the study seeks to propose cost-saving benefits by encouraging informed decisions in structural design and material selection to promote environmental sustainability in construction practices.

## 2. Methods and Material Properties

The research methodology can be briefly outlined in the following phases.

- Phase 1 (green concrete): Define concrete mixture models and gather environmental and cost data for the three types.
- Phase 2 (BIM): Conduct architectural modeling for a ten-story concrete dormitory building using Revit.
- Phase 3 (AxisVM): Develop a structural model for the ten-story RC dormitory building using AxisVM.
- Phase 4 (sustainable selection): Use the structural model to obtain and analyze the results pertaining to sustainability impacts, including embodied $CO_2$ and cost.

The cement utilized in this study is CEM I 52.5 N, distinguished by a relative density of 3.12 g/cm$^3$. Moreover, in recycled concrete productions, silica fume and fly ash were utilized as substitutes for cement at rates of 12% and 20%, respectively. Their relative densities were 2.23 and 2.45 g/cm$^3$. Three distinct types of local coarse natural river quartz aggregates, all of which acquired the highest size of 16 mm, were incorporated. The recycled concrete aggregate was derived from crushed concrete composed of natural aggregates after a 90-day period. The sand used in each mixture had the highest size of 4 mm (685 kg/m$^3$). Tap water was used for the mixing process (145 kg/m$^3$). However, to improve the plastic consistency of concrete production, a superplasticizer, Sika ViscoCrete-5-500, was used in specific proportions. This superplasticizer, with a relative density of 1.07 g/cm$^3$, was applied in a percentage ratio ranging from 1.2% to 1.6%. Figure 1 shows all the steps of the paper.

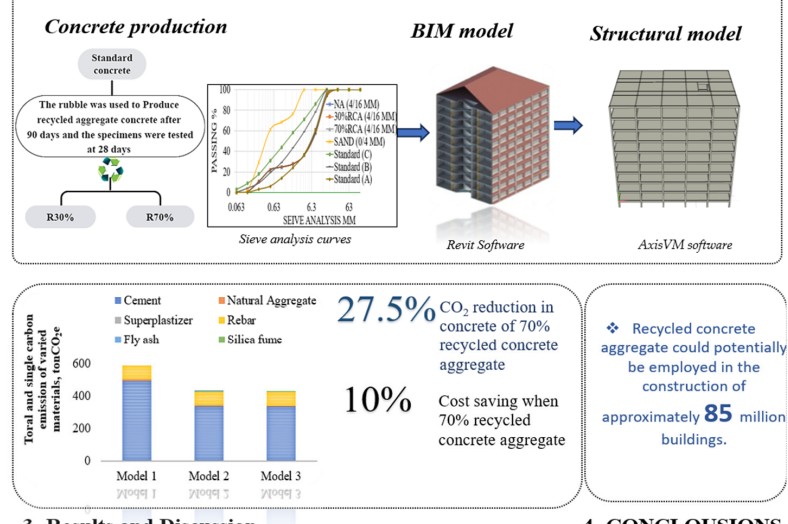

**Figure 1.** A graphical representation of the paper stages.

### 2.1. Architectural Modeling (BIM)

BIM emerges as a highly effective means to visualize structures and models prior to actual construction [23]. BIM facilitates collaboration and interoperability, enhancing project efficiency throughout its life cycle, from planning to demolition. In summary, the construction industry is the primary contributor to global $CO_2$ emissions, with cement alone responsible for 5% of all $CO_2$ emissions [24]. However, engineers utilizing BIM and stakeholders are increasingly recognizing the essential role of their sector in reducing project expenses, material requirements, and waste.

To design, develop, and display a three-dimensional virtual model of a ten-story dormitory structure, Autodesk Revit was selected as the technology for this work. One building was designed using Revit software, incorporating different concrete compositions: standard concrete, concrete with 70% recycled aggregate, and concrete with 30% recycled aggregate. Figure 2 shows the 3D and plan views of the architectural model using Autodesk Revit. The above-established methodology was applied to these designed concrete buildings, resulting in results that met the predetermined standard [25]. Revit was selected primarily for its ability to generate drawings and models that highlight the fundamental principles of BIM. Subsequently, the concrete and reinforcement constitution of structural components from structural design using the AxisVM program were applied. This specific software was used for the examination of RC elements, providing information on individual volumes of concrete and reinforcement. The obtained volume data serve as a crucial parameter for analyzing the $CO_2$ emissions associated with the structural elements of the building under consideration.

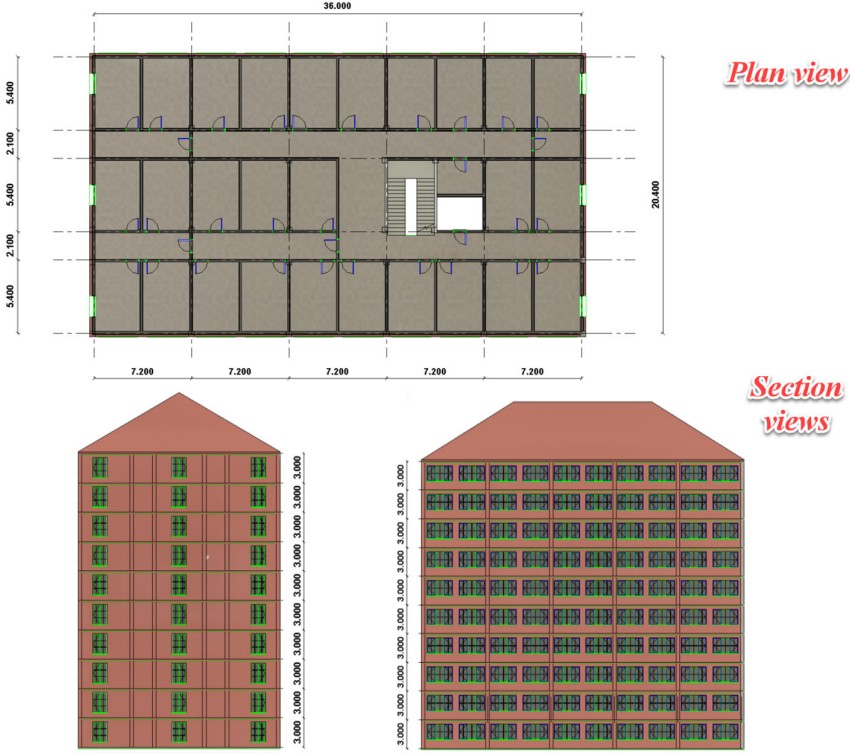

**Figure 2.** Plan and 3D section views of the designed building.

### 2.2. Structural Modeling and Analysis

The focus of this study is on studying a dormitory building in Győr, Hungary, designed with RC. When designing structures to withstand seismic forces, it is imperative to take into account the seismicity of the site. The seismic hazard map of Hungary, depicted in Figure 3, reveals that the seismicity of Hungary is categorized into five zones, each associated with Peak Ground Acceleration (PGA) values ranging from 0.08 g to 0.15 g. These PGA values indicate that Hungary is located in a region characterized by moderate seismicity. During

the seismic design process, a PGA value of 0.12 g is taken into account, which represents the seismicity of the site, since the dormitory building was located in Győr.

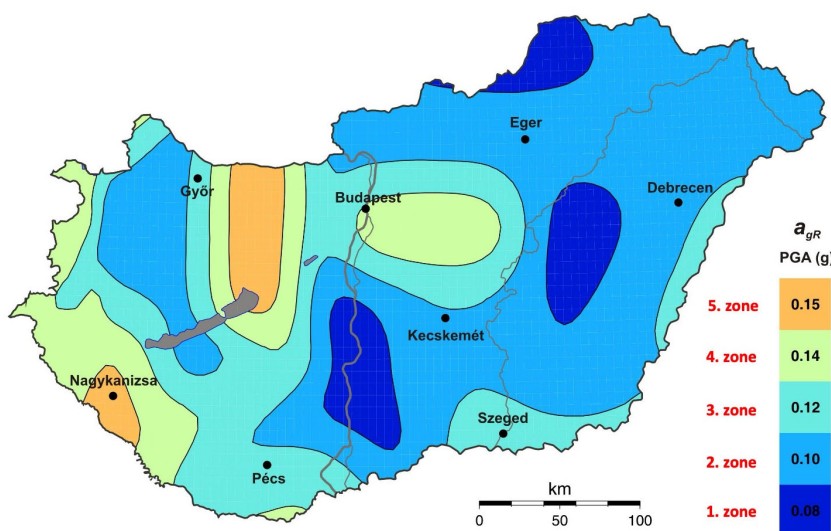

**Figure 3.** Seismic hazard map used in Hungary to determine peak ground acceleration [26].

In addition to site seismicity, another crucial parameter to address in the design process is the characterization of site soil properties. The site soil properties include six soil types which are classified according to the average shear wave velocity of the site: Type A (>800 m/s), Type B (360–800 m/s), Type C (180–360 m/s), Type D (<180 m/s), and Type E, which includes S1 (<100 m/s) and S2 [27]. The Győr micro zonation map, derived from these investigations, is presented in Figure 4. Upon examination of the figure, it becomes evident that, despite the presence of some locations with B-type soil, the predominant soil type in Győr is classified as C. Cohesive soils classified as Type B are distinguished by their unconfined compressive strength, which is greater than 48 kPa but less than 144 kPa [27]. On the contrary, cohesive soils with an unconfined compressive strength of less than 48 kPa are known as Type C soils [27]. Consequently, for the purposes of this investigation, the design of the building has taken into account the prevailing soil type of C.

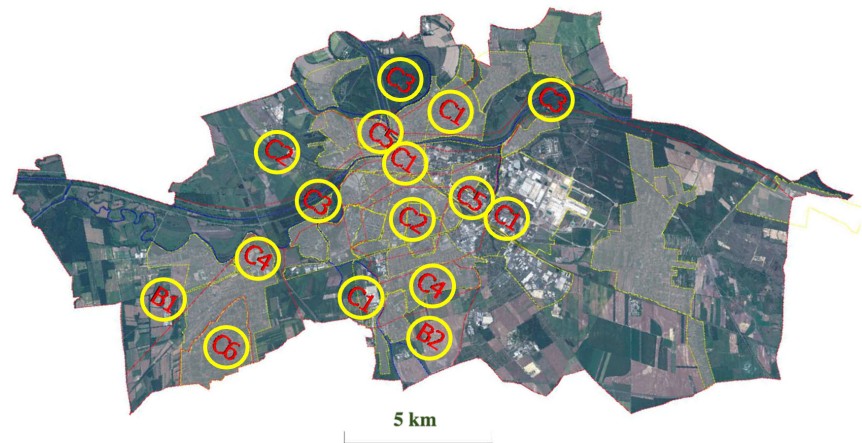

**Figure 4.** Győr soil classification map [28].

The design response spectrum for the building has been established by integrating the soil properties and PGA of the site. The considered values include $a_{gr}$, which is the ground acceleration, set at 1.2 m/s$^2$, the soil type designated as C, and an assigned building importance factor of 1.2, per Eurocode 8 [27]. The vertical red lines in the design response spectrum correspond to the periods of different modes, as shown in Figure 5.

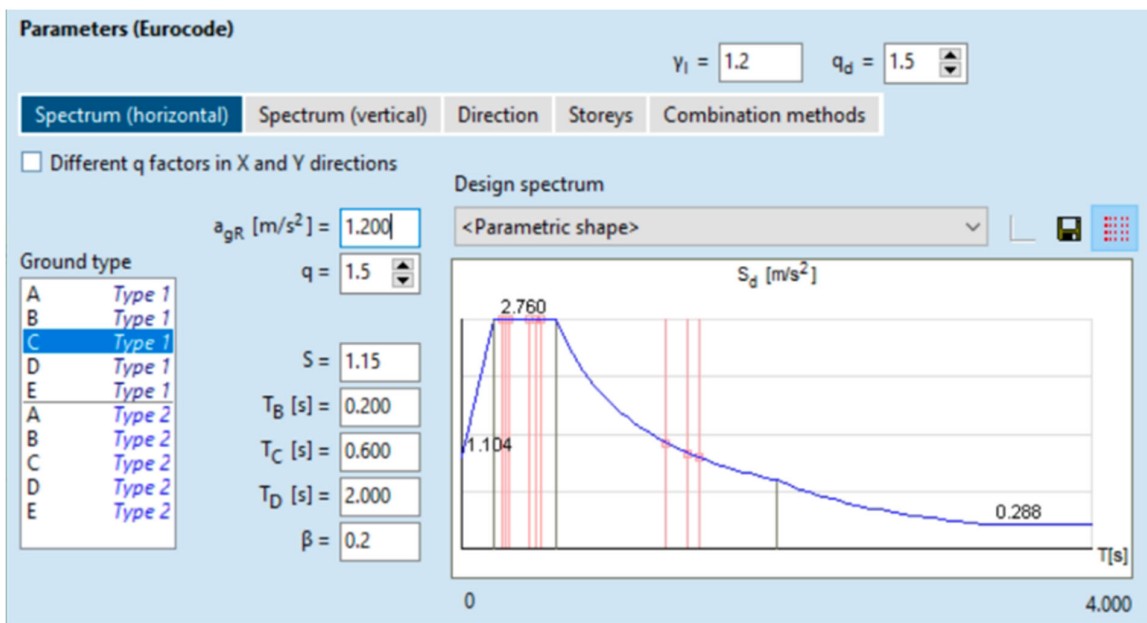

**Figure 5.** Design response spectrum based on building importance, site seismicity, and soil properties for seismic design.

The building designed in this research is an RC structure comprising a ground floor and nine additional floors. The structural system adopted for the building is a moment-resisting frame. The planned dimensions of the building measure 36 m by 20.4 m, with a story height of 3 m. The building exhibits a rectangular shape on the plan, and there are no discernible vertical or plan irregularities. In the construction of the building, C55/67 concrete was utilized. The properties of the concrete material include a modulus of elasticity of 38.2 MPa and a Poisson ratio of 0.2. The reinforcing steel grade used in this study is B500B. The elastic modulus of reinforcement steel is 200 GPa, the Poisson's ratio is 0.29, and the yield strength is 435 MPa. The designed structure comprises 38 columns, each measuring $35 \times 35$ cm, and 390 beams, with beam dimensions are $30 \times 50$ cm. The slab is modeled as a shell element with a thickness of 15 cm.

*2.3. Environmental Impact and Cost Assessment*

The volume consumption of carbon emissions was the only environmental criterion considered to assess the environmental performance of studied models constructed from RC buildings. As mentioned above, cement is the main component that produced the huge amount of $CO_2$ emissions, which varied from one building to another, as fly ash and silica fume were used in place of cement in our study. Furthermore, the natural aggregate was deemed to be a competent variable, as it changed based on the ratio of a recycled concrete aggregate, and superplasticizer was found to be more competent as the ratio of the recycled concrete aggregate increased. Table 1 shows the percentage of the cement, natural aggregate, and superplasticizer needed for each designed model.

**Table 1.** The percentage of varied materials used in each model.

| Model | Concrete Type | Fly Ash (%) | Silica Fume (%) | Cement (%) | Natural Aggregate (%) | Recycled Aggregate (%) | Superplasticizer (%) |
|---|---|---|---|---|---|---|---|
| Model 1 | Standard concrete | 0 | 0 | 15 | 50.37 | 0 | 0.180 |
| Model 2 | 30% recycled aggregate concrete | 20 | 12 | 10.2 | 35.37 | 15.17 | 0.225 |
| Model 3 | 70% recycled aggregate concrete | 20 | 12 | 10.2 | 15.17 | 35.37 | 0.240 |

However, the $CO_2$ values of the various materials used in concrete-designed models could be summed up using Equation (1) [29].

$$CO_2, \text{total} = \Sigma \, (W_i \times CO_{2,j}) \tag{1}$$

$W_i$ is the total amount of each j material in kg, $CO_2$, j is the $CO_2$ emission value of each j material in $kgCO_2/kg$, and $CO_2$, total is the total $CO_2$ emission of concrete in $kgCO_2/kg$.

In this study, the chosen environmental indicator was based on a subset of $CO_2$ emission estimates sourced from the literature, specifically adapted for application in Europe, with a focus on Hungary. The assigned $CO_2$ values for cement, natural aggregate, superplasticizer, and rebar were approximately 0.7667 $kgCO_2/kg$, 0.0029 $kgCO_2/kg$, 0.25 $kgCO_2/kg$, and 0.93 $kgCO_2/kg$.

For the purpose of coordinating and using the cost analysis approach, several rules and regulations have recently been developed [30]. Still, there is no specific approach for figuring out expenses in the construction field. But it only provides a financial view of a product's life cycle, while the life cycle provides ecological details on emissions that contribute to climate change and pollution. However, to complete the cost analysis and provide a basis for the calculations, the viewpoint integrated by one or more of the stakeholders participating in the product life cycle must be fixed. In our work, the cost of recycled coarse aggregate (no transportation) was produced in our laboratory, and the cost was 0.004 euro/kg. Meanwhile, cement, coarse/fine aggregate, fly ash, silica fume, superplasticizer, and rebar were 0.2 euro/kg, 0.01 Euro/kg, 0.172 Euro/kg, 0.06 Euro/kg, 5.25 Euro/kg, and 0.22 Euro/kg, respectively. The international wholesale price (https://Alibaba.com) and applicable transportation expenses are the basis for all materials used. According to the suppliers, the total cost of each concrete mixture (excluding transportation expenses) was calculated by combining the prices of each component required to manufacture one cubic meter of concrete. Additionally, the transportation cost was 6.3 Euro/20 km.

## 3. Results and Discussion

This section encompasses (I) the utilization of materials (concrete and reinforcement) in the design of the RC building under consideration, (II) an evaluation of the EI of these materials, and (III) a discussion on earthquake-induced waste management and the recycling effects on the environment, drawing from the findings of the case study.

### 3.1. Analyzing Material Usage: Concrete and Reinforcement

This section delves into the material usage, with a focus on concrete and reinforcement, in the seismic design of the RC dormitory building, as shown in Table 2. Based on the experimental findings, a range of concrete strengths was observed ranging from 55.4 to 66.3 MPa, in agreement with the C55/67 concrete classification. In the concrete strength of analysis, the 55 MPa was settled upon as it serves as a representative value for both standard concrete and recycled aggregate concrete models.

**Table 2.** Material used to design the building.

|  | Parts | Volume [m$^3$] | Weight [kg] |
|---|---|---|---|
| Concrete | Column | 521.100 | 1,302,750.052 |
|  | Beam | 139.650 | 349,124.988 |
|  | Slab | 1057.860 | 2,644,650.105 |
|  | Total | 1718.610 | 4,296,525.14 |
| Reinforcement | Slab rebar | 1057.888 | 59,475.531 |
|  | Beam rebar | 520.557 | 16,956.845 |
|  | Beam stirrups | 23.734 | 28,291.895 |
|  | Column rebar | 274.152 | 9768.521 |
|  | Column stirrups | 98.999 | 3602.926 |
|  | Total | 1715.526 | 90,114.055 |

The construction project involved the utilization of over 4296 tons of concrete and 90 tons of steel. The distribution of materials, encompassing both concrete and reinforcement, is illustrated in Figure 6. Figure 6a demonstrates that 61.6% of total concrete was allocated to slabs, 30.3% to columns, and 8.1% to beams. Figure 6b indicates that 50.4% of the total reinforcement was designated for slab reinforcement, 24% for beam stirrups, 14.4% for beam rebars, 8.3% for column rebars, and 3.1% for column stirrups. Figure 6c represents that 97.9% of the total material employed in the building's design is concrete, with the remaining 2.1% constituting the reinforcement.

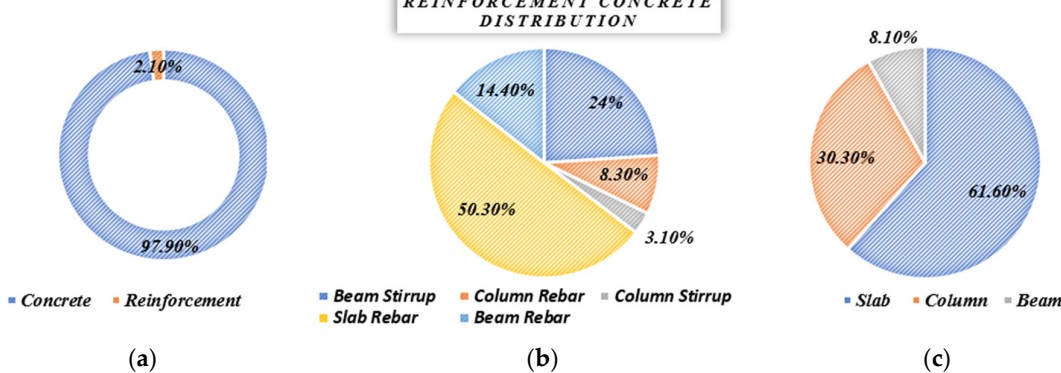

**Figure 6.** Distribution of concrete and reinforcement across structural components and system: (**a**) examines concrete distribution across structural elements, (**b**) evaluates reinforcement material distribution by type, and (**c**) contrasts the total weight of concrete versus reinforcement.

### 3.2. Assessing Carbon Emissions across Structural Models

Figure 7 displays the proportion of the materials of each model. The highest ratio is for natural aggregate, which represents 75% of the concrete components in the standard concrete model. However, cement up to 22% has the second highest ratio. From this point on, the use of waste materials is worthwhile, particularly after the earthquake.

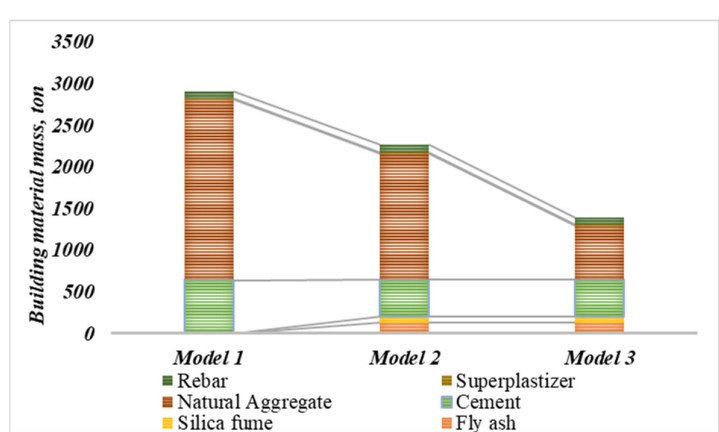

**Figure 7.** Building material mass.

By utilizing the previously mentioned Equation (1) for computations, Model 1 revealed an overall carbon emission of 586.138 tons of $CO_2$. Model 2 states that a total of 426.907 tons of $CO_2$ was released. In contrast, model 3 displayed the value of 424.5511 tons of $CO_2$ as total carbon emissions. The study covers carbon emissions from construction to end-of-life, with an expected 50-year lifetime. The raw material stage exhibits the highest carbon emissions in Model 1. However, Model 2 ranks as the second highest in carbon emissions compared to standard concrete. Detailed comparisons of embodied carbon emissions (in tons $CO_2$e) and the mass of materials from the three design model options, considering both virgin and supplementary cementitious materials, are presented in Figure 8. Notably, incorporating a high dose of recycled concrete aggregate (up to 70%) results in a 27.5%

reduction in carbon emissions compared to standard concrete, representing that low carbon dioxide reductions are attributed to the substitution of natural aggregates.

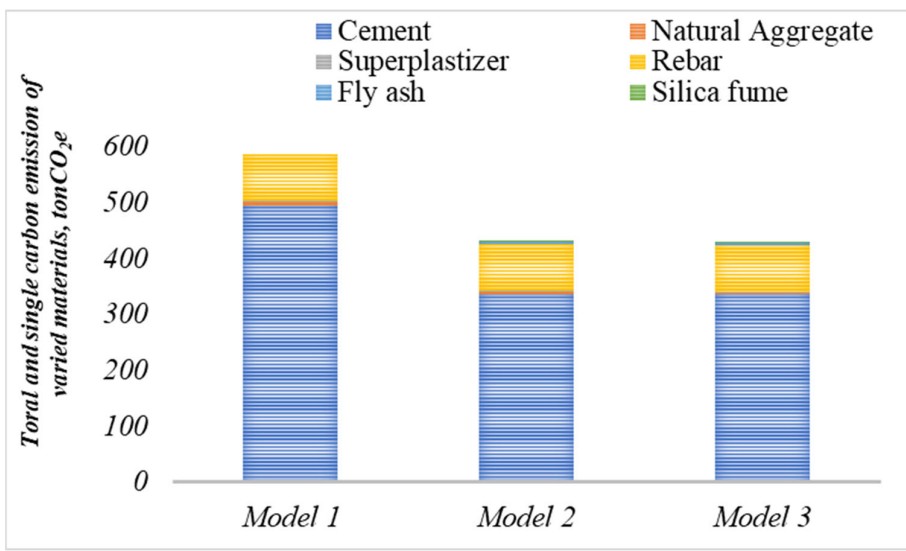

**Figure 8.** Total and single carbon emission of models.

*3.3. Cost Evaluation*

For the cost of materials, two scenarios were applied in this study:

(1)    Without transportation: Lower building material prices were achieved by using recycled concrete aggregate on one side and supplemental materials on the other, as confirmed by many investigations, [31]. However, since the recycled concrete aggregate used in this study was crushed physically in the lab of the university, there were no costs associated with transportation. Furthermore, there were no aggregate crushing costs, and a low price of the material. As a consequence, model 3 and model 2 have the lowest pricing compared to model 1, having fallen by 6.9% and 9.13%, respectively. However, the estimated cost of all models is shown in Figure 9.

(2)    With transportation: To illustrate the impact of transportation distance on the efficacy of natural aggregate substitution, the following plans are applied: (1) transporting both natural aggregate and recycled concrete aggregate over 100 km, (2) transporting natural aggregate 100 km and recycled concrete aggregate 50 km, with a replacement ratio of 30%, and (3) transporting natural aggregate 100 km and recycled concrete aggregate 50 km with a replacement ratio of 70%. These distances are selected based on an alternative plan that assumes closer distances for transporting recycled concrete aggregate due to the proximity of the sites. It is worth noting that obtaining recycled concrete aggregate involves gathering materials from fragmented locations, making the determination of the transport distance complex. Nevertheless, the issue of distance is crucial, as the transportation of large quantities of aggregate necessitates significant resources. Additionally, in case of an earthquake, there will surely only be a need to transport equipment and personnel to manage the recycling process at the location of the earthquake and the subsequent construction work. Thus, the first scenario can be the correct one for our laboratory work.

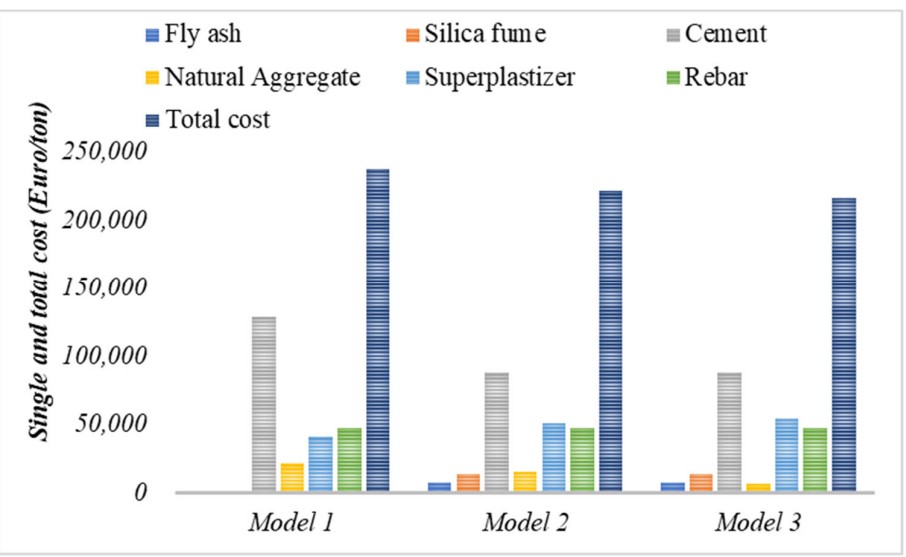

**Figure 9.** The costs of model compositions and the total costs of the models.

*3.4. Earthquake-Induced Waste Management and Recycling Impact Analysis*

After the Türkiye–Syria earthquake, an estimated 450,920 million tons of disaster debris consisted of concrete debris, representing a significant portion [6]. Using the percentage of concrete composition in the total earthquake-induced waste, the amount of concrete waste is determined, as outlined in Table 3. The reuse of earthquake-induced structural waste can help reduce the environmental impact and mitigate the demand for raw materials. However, to calculate waste management, a percentage of the recycled aggregate was used in the composition of the concrete material to estimate the quantity of the total recycled aggregate required for the building's construction. Although previous studies [15,29,31] have delved into material development utilizing recycled aggregate concrete, this research advances the field by examining the implications of earthquake-induced waste for new construction projects, as highlighted in the literature [32]. Given the amount of concrete waste generated by the earthquake, it becomes apparent that a recycled concrete aggregate could potentially be employed in the construction of approximately 85 million buildings. The utilization of such waste not only contributes to improving waste classification efforts, but also plays a crucial role in mitigating the long-lasting effects of disasters [12]. Table 3 provides an overview of the possibilities of concrete use and recycling following the earthquake, demonstrating the amount of waste produced and the viability of using recycled materials in building projects.

**Table 3.** Summary of concrete usage and recycling potential post-earthquake concrete waste [6].

| Parameter | Amount | Unit |
|---|---|---|
| Total concrete waste from the Türkiye–Syria Earthquake | 129,025.52 | million tons |
| Concrete utilized in the RC building design | 4296.89 | tons |
| Recycled concrete aggregate utilized in the RC building | 1519.89 | tons |
| Proportion of recycled aggregate in total | 35 | % |
| Estimated number of buildings constructed using concrete waste | ~85 | million |

The findings of this study underscore the feasibility of repurposing structural waste, such as concrete. Structural waste should be actively directed towards construction applications, and any residual materials, particularly concrete remnants, that cannot be directly reused can be repurposed as infill material.

The extensive recycling of concrete waste significantly contributes to waste reduction, thus mitigating the adverse impacts of waste on human settlements and the environment. Sustainable development within the built environment necessitates considerations ranging

from the design phase to end-of-life scenarios. Integrating earthquake-induced waste management into discussions about recycling concrete emphasizes the importance of long-term waste management practices, especially in seismically active areas. Communities can reduce the EI of disasters by repurposing earthquake-related debris for construction, while also promoting resource efficiency and resilience in the built environment. The sustainable management of disaster-induced waste extends beyond cradle-to-cradle approaches by fostering a circular economy mindset, emphasizing reuse, recycling, or repurposing over simple disposal. This strategy promotes resilience, disaster preparedness, and avoids using lands for waste by fostering adaptable and robust systems.

## 4. Conclusions and Future Work

Recent seismic events have highlighted the considerable waste generated by building destruction during severe earthquakes, underscoring the urgent need for effective disaster-induced waste disposal and recycling initiatives. Analytically, a comprehensive 3D architectural Revit building is developed, detailing the number of rooms and elements on each floor, and seismic analysis is conducted on the RC dormitory building, which is located in Győr, Hungary, using AxisVM software. The primary criteria for evaluation include the EIs and cost effects. Furthermore, the research explores the integration of recycled concrete aggregates as part of earthquake-induced structural waste management strategies in the construction of new buildings. The findings can be summarized as follows.

The research thoroughly generated and examined a realistic building model using AxisVM and Revit software, finding a large percentage (35% and 1519.89 tons) of concrete waste that can be included in the construction.

The building's carbon emissions have been reduced by 161.587 tons through the design models.

Considering the total concrete waste from the Türkiye–Syria earthquake, it is evident that waste construction could potentially be used to construct approximately 85 million buildings.

An impressive cost-saving advantage of approximately 10% became more evident when supplementary materials and recycled concrete aggregate were used instead of traditional cement and natural aggregate.

Promoting sustainability and incorporating recycled materials into construction enhances the resilience and efficiency of the built environment while also mitigating adverse environmental and climate change impacts. Future considerations include the following:

- The emphasis on the utilization of BIM with recycled concrete aggregates in construction projects can serve as a crucial element within sustainable development endeavors, providing earthquake-resistant and environmentally friendly buildings.
- Investigating the closed-loop recycling of aggregate concrete presents an intriguing avenue for exploring LCA of this concrete type.
- Manual calculations (based on Equation (1)) were used to estimate the carbon emissions for the three models in this study. However, future research could broaden its scope to include evaluations of additional environmental factors, including human health, resource consumption, climate impact, and ecosystem integrity. This expansion could utilize software tools such as SimaPro, OpenLCA, and GaBi.
- It is advisable to expand the scope of sustainability considerations in waste management by incorporating social impacts, particularly emphasizing recycling and reusing debris.
- It is recommended to develop comprehensive disaster-induced debris management plans that are adaptable to various regions.
- A comparative review of disaster debris management efforts within the industry, along with their limitations and research findings from the literature, is essential to guide future research perspectives.

**Author Contributions:** N.B. and M.S.: Conceptualization; methodology; software; validation; formal analysis; investigation; resources; data curation; writing—original draft preparation; writing—review and editing; visualization; supervision; project administration. All authors have read and agreed to the published version of the manuscript.

**Funding:** This research received no external funding.

**Data Availability Statement:** All data supporting the findings of this study are included within the paper.

**Acknowledgments:** Many thanks to the laboratory workers and the previously mentioned authors at Széchenyi István University.

**Conflicts of Interest:** The authors declare no conflicts of interest.

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
