# Peer review of "Earthquake-Induced Waste Repurposing: A Sustainable Solution for Post-Earthquake Debris Management in Urban Construction"

_buildings, doi:10.3390/buildings14040948_

Round 1

Reviewer 1 Report

Comments and Suggestions for Authors

The authors did a good job of studying “Earthquake-Induced Waste Repurposing: A Sustainable Solution for Post-earthquake Debris Management in Urban Construction”. I suggest authors to carryout minor revisions / clarifications in order to enhance the manuscript quality.

1.    In section 1, I suggest authors to indicate the key challenges in post-earthquake debris management in urban construction in Introduction part.

2.    In figure 1, I recommend authors to incorporate dimensions of plan model.

3.    I suggest authors to include discussions regarding further research or development needed to improve and optimize the proposed solution in conclusion part.

4.    I suggest authors to check for grammar corrections in manuscript.

Author Response

Firstly, we would like to express our gratitude to the Reviewer for his/her time and significant feedback. The modifications and/or additions made in response to the comments below are explained one by one.

The authors did a good job of studying “Earthquake-Induced Waste Repurposing: A Sustainable Solution for Post-earthquake Debris Management in Urban Construction”. I suggest authors to carryout minor revisions / clarifications in order to enhance the manuscript quality.

  1. In section 1, I suggest authors to indicate the key challenges in post-earthquake debris management in urban construction in Introduction part.

Response:

Thank you for your comment. Another paragraph has been included in the introduction section to offer clarification specifically on waste management, focusing on earthquake-induced debris management, in response to the reviewer's feedback.

Notes/actions:

1. Introduction – page 2

  1. In figure 1, I recommend authors to incorporate dimensions of plan model.

Response:

Done. The figure has been updated to clearly reflect the dimensions of the building, as per the recommendation.

Notes/actions:

Figure 2 – Page 5

  1. I suggest authors to include discussions regarding further research or development needed to improve and optimize the proposed solution in conclusion part.

Response:

Further research directions have been incorporated into Section 5: Conclusion and Future Work, in accordance with the reviewer's recommendation.

Notes/actions:

5. Conclusion and future work – Pages 12 and 13

  1. I suggest authors to check for grammar corrections in manuscript.

Response:

The grammar of the paper has been thoroughly checked.

The adjustments and additions made in this study are shown in green text in the paper, making it easy to follow all of the changes.

Reviewer 2 Report

Comments and Suggestions for Authors

This study introduces an approach, using three types of concrete that have different ratios of recycled aggregate, fly ash and silica fume from the literature for analyzing designed models (dormitory building located in Gyor, Hungary- an RC moment frame structure) by incorporating Revit software for architectural design and AxisVM for structural design, along with LCA (Life Cycle Assessment).Where, the primary criteria for evaluation include environmental impact (EIs) and cost effects. Furthermore, the research explores the integration of recycled concrete aggregates as part of earthquake-induced structural waste management strategies in the construction of new buildings. There are the following comments:

1-    There are many researches about recycled aggregates, so it is required to improve the introduction through referring to these researches related to the topic of recycled aggregates. For example, you can see this paper (Flexural performance of reinforced concrete beams made by using recycled block aggregates and fibers)

2-    It is required to mention the meaning of symbol LCA at first time appear in the manuscript

3-    Clarify the meaning of the soil type B and the soil type C

4-    The figures are not clear enough

5-    Add references for Table 3

6-    You mentioned (However, since the recycled concrete aggregate used in this study was physically crushed in the lab of the university, there were no expenses associated with transportation) in rows (246 and 247) of page 8. It is not practical to crush huge quantities of waste concrete in lab.  

Author Response

1       Reviewer – 2

Firstly, we would like to express our gratitude to the Reviewer for his/her time and significant feedback. The modifications and/or additions made in response to the comments below are explained one by one.

This study introduces an approach, using three types of concrete that have different ratios of recycled aggregate, fly ash and silica fume from the literature for analyzing designed models (dormitory building located in Gyor, Hungary- an RC moment frame structure) by incorporating Revit software for architectural design and AxisVM for structural design, along with LCA (Life Cycle Assessment).Where, the primary criteria for evaluation include environmental impact (EIs) and cost effects. Furthermore, the research explores the integration of recycled concrete aggregates as part of earthquake-induced structural waste management strategies in the construction of new buildings. There are the following comments:

1-    There are many researches about recycled aggregates, so it is required to improve the introduction through referring to these researches related to the topic of recycled aggregates. For example, you can see this paper (Flexural performance of reinforced concrete beams made by using recycled block aggregates and fibers)

Response:

As recommended by the reviewer, further explanation has been incorporated, and the section on recycled aggregate was extended.

Notes/actions:

1. Introduction – Page 2

2-    It is required to mention the meaning of symbol LCA at first time appear in the manuscript

Response:

The extended version of the Life Cycle Assessment (LCA) has been incorporated into the paper, as suggested by the reviewer.

Notes/actions:

1. Introduction – Page 2

3-    Clarify the meaning of the soil type B and the soil type C

Response:

In response to the reviewer's comments, further explanation regarding soil types according to Eurocode has been integrated into the paragraph preceding Figure 3. This includes elaboration on soil types B and C for enhanced clarity and understanding.

Notes/actions:

Section 2.2. Structural Modelling and Analysis – Page 6

4-    The figures are not clear enough

Response:

We have improved the quality and visibility of Figures 2, 3, 4, and 6 in response to the reviewer's feedback.

5-    Add references for Table 3

Response:

In response to the reviewer's comment, a corresponding reference was added to Table 3.

Notes/actions:

Section 3.4. Earthquake-Induced Waste Management and Recycling Impact Analysis – Page 11

6-    You mentioned (However, since the recycled concrete aggregate used in this study was physically crushed in the lab of the university, there were no expenses associated with transportation) in rows (246 and 247) of page 8. It is not practical to crush huge quantities of waste concrete in lab.

Response:

Thank you for highlighting this point. Two scenarios are added to our study. In fact, it matters in real conditions, especially if environmental and cost factors are taken into account. But in case of earthquake, there will surely be only a need to transport equipment and personnel to manage the recycling process in the location of the earthquake and the subsequent construction work. Thus, the first scenario can be the correct one for our laboratory work. 

Notes/actions:

Section 3.3

The adjustments and additions made in this study are shown in green text in the paper, making it easy to follow all of the changes.

Reviewer 3 Report

Comments and Suggestions for Authors

Review Report:

The paper titled as “Earthquake-Induced Waste Repurposing: A Sustainable Solution for Post earthquake Debris Management in Urban Construction”

shows some innovative results and concepts that shows some promising results especially the post-earthquake waste utilization.

The results could be interesting, however at this stage the manuscript does not present clear results and discussion on the topic.

Life cycle assessment of the mentioned materials is suggested to further understand the positive influence of the recycled materials acquired from the earthquake debris. Besides, some material testing (mechanical/microstructural) is recommended.

BIM technique is an interesting way to visualize the structure. However, being a structural engineer or even construction manager this aspect is not enough to justify the potential use in the construction industry.

At this stage the paper seems to be a lab report which is not suitable for publication in this prestigious journal and I suggest a rejection.

Author Response

1       Reviewer – 3

Initially, we would like to thank the Reviewer for his/her time and significant comments. Changes and/or additions made in the light of the comments made are explained separately under each of the comments listed below.

Review Report:

The paper titled as “Earthquake-Induced Waste Repurposing: A Sustainable Solution for Post earthquake Debris Management in Urban Construction” shows some innovative results and concepts that shows some promising results especially the post-earthquake waste utilization.

The results could be interesting, however at this stage the manuscript does not present clear results and discussion on the topic.

Response:

The Cost Evaluation section has been expanded to incorporate the reviewer's feedback. Additionally, a new figure (Figure 9) has been included in this section to provide a comparative visual assessment of three models. Furthermore, further explanation has been added to elucidate the importance of sustainable waste management.

Notes/actions:

Section 3.3. Cost evaluation – Pages 10, 11

3.4. Earthquake-Induced Waste Management and Recycling Impact Analysis – Page 12

Life cycle assessment of the mentioned materials is suggested to further understand the positive influence of the recycled materials acquired from the earthquake debris. Besides, some material testing (mechanical/microstructural) is recommended.

Response:

We appreciate your feedback. We have completed the mechanical testing, and it has been published in a different journal. But unlike other studies, ours is analytical research that combines BIM and LCA. To analyze the impact of recycled aggregate in actual buildings, we exclusively employed the strength of the concrete under investigation as an input into the AxisVM software.

BIM technique is an interesting way to visualize the structure. However, being a structural engineer or even construction manager this aspect is not enough to justify the potential use in the construction industry.

Response:

We appreciate your comment. However, BIM offers more than visualization; it enhances collaboration, improves decision-making, streamlines project management, reduces costs, supports lifecycle management, and ensures regulatory compliance, making it indispensable in the construction industry. In our study we only need the details of the studied building and to show reliability of our work.

At this stage the paper seems to be a lab report which is not suitable for publication in this prestigious journal and I suggest a rejection.

The adjustments and additions made in this study are shown in green text in the paper, making it easy to follow all of the changes.

Reviewer 4 Report

Comments and Suggestions for Authors

The manuscript is about the Earthquake-Induced Waste Repurposing which is a Sustainable Solution for Post-earthquake Debris Management in Urban Construction. Here are some comments

1 The soil types B C are the ones according to Eurocode 8? Provide some material variables that come along this classification.

2 Provide cost estimation for all models. That would enlighten your methods qualities. This refers to chapter 3.

3 What is the reinforcement steel quality? Is it B500C?

4 Young modulus of concrete and reinforcement should also mentioned alongside with Poisson ratio

5 Provide reference of the estimations of CO2 emissions for all materials

Comments on the Quality of English Language

NA

Author Response

1       Reviewer – 4

Firstly, we would like to express our gratitude to the Reviewer for his/her time and significant feedback. The modifications and/or additions made in response to the comments below are explained one by one.

The manuscript is about the Earthquake-Induced Waste Repurposing which is a Sustainable Solution for Post-earthquake Debris Management in Urban Construction. Here are some comments

1 The soil types B C are the ones according to Eurocode 8? Provide some material variables that come along this classification.

Response:

In response to the reviewer's comments, further explanation regarding soil types according to Eurocode has been integrated into the paragraph preceding Figure 4. This includes elaboration on soil types B and C for enhanced clarity and understanding.

Notes/actions:

Section 2.2. Structural Modelling and Analysis – Page 6

2 Provide cost estimation for all models. That would enlighten your methods qualities. This refers to chapter 3.

Response:

The costs of model compositions and the total costs of the models are illustrated in Figure 9. A corresponding explanation is provided preceding Figure 9 for clarity and context.

Notes/actions:

Section 3.3. Cost Evaluation – Pages 10 and 11

3 What is the reinforcement steel quality? Is it B500C?

Response:

In response to the reviewer's comment, information regarding the B500B reinforcement steel has been incorporated into the paper in the paragraph after Figure 5.

Notes/actions:

Section 2.2. Structural Modelling and Analysis – Page 7

4 Young modulus of concrete and reinforcement should also mentioned alongside with Poisson ratio.

Response:

To respond to the reviewer's comment, the material properties of concrete and reinforcement are described in the paragraph following Figure 5.

Notes/actions:

Section 2.2. Structural Modelling and Analysis – Page 7

5 Provide reference of the estimations of CO2 emissions for all materials.

Response:

Thank you for your comment. Reference 29 provides the estimated CO2 emissions for all materials used in the study.

Notes/actions:

Section 2.3. Environmental Impact and Cost Assessment – Page 8

The adjustments and additions made in this study are shown in green text in the paper, making it easy to follow all of the changes.

Round 2

Reviewer 3 Report

Comments and Suggestions for Authors

the paper has been improved. I recommend to accept it for publication